# Deformation Behavior and Microstructure of 6061 Aluminum Alloy Processed by Severe Plastic Deformation Using Biaxial Alternate Forging

**DOI:** 10.3390/ma17050968

**Published:** 2024-02-20

**Authors:** Seong-Ho Ha, Young-Chul Shin

**Affiliations:** Korea Institute of Industrial Technology, Incheon 21999, Republic of Korea; shha@kitech.re.kr

**Keywords:** biaxial alternate forging, severe plastic deformation, FEM, AA6061, mechanical property

## Abstract

The deformation behavior and microstructure of 6061 aluminum alloy processed by severe plastic deformation (SPD) using biaxial alternate forging that can evaluate the forming limit and mechanical properties of alloys, simultaneously, were investigated in this study. A finite element (FE) analysis on the biaxial alternating forging process, considering the strain-hardening coefficient and forging pass of the material, was conducted. When the strain-hardening coefficient is 0, an average effective strain of 440% was found within a diameter of 4 mm in the core of the workpiece after eight passes, while it was 300% at the same pass number when the strain-hardening coefficient was 0.2. The average effective strain estimated from the FE analysis was about 264% after eight passes of forging, which is considered to be a level of SPD that significantly exceeds the elongation of the raw material. As a result of the tensile test according to the forging pass, after two passes, the strength of the material could be gradually improved without significant degradation of elongation. Even though a large strain of 264% was found after eight passes were applied, deformed grains and twins with no recrystallized structure in optical microstructures with different forging passes were found.

## 1. Introduction

Severe plastic deformation (SPD) involves significantly large strains with a complex stress state or high shear, resulting in a high dislocation density [1]. The main purpose of the SPD process is to produce high-strength and lightweight components with environmental harmony [1]. In the typical metalworking processes such as rolling, forging, and extrusion, the plastic strain imposed is generally less than approximately 2.0. When multipass rolling, drawing, and extrusion are conducted up to a strain greater than 2.0, the thickness of parts becomes very thin and is not suitable to be used for structural parts [1]. In order to impose a significantly large strain on the bulk metals while keeping the shape, various SPD processes on Al alloys have been examined. 

Zhang et al. [2] investigated the effect of SPD on Al alloy sheets with enhanced precipitation. In their report, a high-frequency shot peening was adopted to apply the SPD on the surface of sheets. They made an SPD layer of 40–70 μm in thickness from the surface. Rogachev et al. [3] studied the as-cast Al–Ca–Mn–Fe alloy processed by the high-pressure torsion (HPT) technique. As a result, an improvement in the strength–ductility balance of the Al–Ca–Mn–Fe alloy was achieved. Manjunath et al. [4] investigated equal channel angular extrusion/pressing to raise the mechanical and physical properties and resistance to wear of the materials. They reported that the hardness and wear resistance of the test materials were improved due to grain refinement. Parimi et al. [5] examined the characteristics of single-phase and two-phase alloys deformed by the multiple channel-die compression method. Extensive grain refinement was obtained, resulting in the formation of nano-sized grains after SPD with a simultaneous increase in flow stress and hardness. Zha et al. [6] investigated Al–7Mg alloy processed by room-temperature equal-channel angular pressing (ECAP) combined with interpass annealing, indicating that an impressive combination of high ductility and high strength was achieved. Zhu et al. [7] studied the microstructural evolution and mechanical properties of Al–Li 2198-T8 alloy processed by friction stir processing (FSP) and HPT. Fine equiaxed grains with different sizes were observed in FSP- and HPT-processed specimens. Rogachev et al. [8] investigated the hardening, structure transformations, and thermal stability of two Al-based eutectic alloys processed by HPT. The HPT led to the formation of a predominantly nanograin structure and the decomposition of eutectic phases. As a result, a significant increase in both the strength and elongation of the alloy was achieved. Naumova et al. [9] examined the phase composition, structure, and properties of an Al–18%Ca alloy containing the Al_4_Ca intermetallic deformed by HPT. They showed the fine structure of the Al_4_Ca intermetallic compound after five turns of HPT. Zhao et al. [10] dealt with the dynamic strain aging of ultrafine-grained Al–Mg alloys with different Mg content deformed by confined channel die pressing at room temperature. The microstructure characterization showed a significant grain refinement and retarded dynamic recovery with an increase in Mg content and SPD. Mohammadi et al. [11] investigated an ultra-SPD realized by HPT to generate a supersaturated solid solution of Al–Zr alloy. The ultra-SPD resulted in the formation of a supersaturated solid solution of approximately 2.9 mass%Zr in Al base metal at room temperature. The increase in Zr concentration in the Al matrix led to the formation of a nanocrystalline structure with a grain size of 73 nm. Moradpour et al. [12] examined finite element modeling and experimental validation of constrained groove pressing for SPD of AA5052 Al–Mg alloy. The newly modified SPD resulted in a more homogenous microstructure and significantly improved the mechanical properties of the alloy. Liu et al. [13] investigated deformation defects in nanostructured an Al–Mg alloy processed by high-pressure torsion. They observed deformation defects such as full and partial dislocations, dipoles, microtwins, and stacking faults using transmission electron microscopy. 

As mentioned above, SPD research on lightweight alloys conducted recently focuses mainly on grain refining to improve the properties of materials and HPT as a method. In our study, a new SPD method using biaxial alternate forging has been examined [14,15]. The biaxial alternate forging method in this study was employed in order to increase strain amounts as desired in cold working. The die system of an octagonal cross-section was designed to maintain the cross-sectional shape of the workpiece regardless of the number of forging passes [14,15]. It enables one to easily insert and take out the workpiece so that a flexible process for the repeated SPD experiment can be carried out. Similar SPD methods by multiple forging have been previously reported [16,17]. Valiev et al. [16] introduced a multiple forging method to form nanostructures in bulk billets. This can be explained by multiple repeats of free forging by varying the axis of the applied strain load. Markushev et al. [17] investigated a multistep isothermal forging of Mg–5.8Zn–0.65Zr alloy, indicating that a homogeneous, recrystallized microstructure with a grain size of 1–2 µm was obtained. From the perspective of multiple forging, the two techniques mentioned above seem to be similar to that in this study. However, the impression-die forging method was applied with the direction limited to two axes in the biaxial alternate forging of this study.

The material to be covered in this study is 6061 aluminum alloy. The 6061 aluminum alloy is one of the very common aluminum wrought alloys produced in various forms to be used for numerous applications because of its excellent mechanical properties, low density, corrosion resistance, and recyclability [18]. Even though it has a high strength-to-weight ratio, the use of aluminum alloys is limited due to their relatively low formability and large spring back at room temperature compared with traditional steel sheets. This paper proposes a feasibility study based on biaxial alternate forging to induce large deformation and upsetting dies with octagonal cross-sections designed to maintain the consistent shape of the workpiece after repetitive operations. The purpose of this study is to examine the deformation behavior and microstructure of the 6061 aluminum alloy processed by SPD using biaxial alternate forging.

## 2. Fundamentals of Biaxial Alternate Forging

Biaxial alternate forging is a method of accumulating continuous deformation in materials through multiforging, and it is designed to facilitate the insertion and extraction of workpieces to speed up the test. Figure 1 shows a schematic of the forging dies and workpieces for biaxial alternate forging. In biaxial alternate forging, as shown in Figure 1a, the rod-shaped workpiece (Φ19 × 108 mm, round 6 mm) is forged into a vertical die with an octagonal cross-sectional cavity shape. Figure 1b provides a three-dimensional view of the die cavity shape, and Figure 2 presents the major dimensions of the die cavity. To accumulate deformation on the material, the workpiece is repeatedly forged by rotating it 90 degrees around the longitudinal direction, as shown in Figure 3. Even after several forgings, the longitudinal cross-section of the workpiece is maintained similarly to that after the forging of one pass. The desired level of strain is controlled through the number of forging passes. The amount of effective strain accumulated in the core of the workpiece per forging pass can be differentiated by adjusting the ratio of vertical and horizontal lengths expressed in D1 (18 mm) and D2 (21 mm), respectively. By increasing the ratio of D2/D1, the effective strain that can be imposed per forging pass can be increased. However, if the ratio of D2/D1 becomes too large, it is difficult to resettle the 90-degree-rotated workpiece to the lower die, so it should be designed at an appropriate ratio. The shapes of the die cavity and both ends of the workpiece were designed to constrain the longitudinal deformation of the workpiece during forging so that the workpiece can undergo plane–strain deformation in the longitudinal direction. This is to increase the reliability of subsequent tensile tests by uniformly maintaining the effective strain in the longitudinal direction of the core of the workpiece during the repetitive forging process. The crevice shape of the dies forms a guide flash on the workpiece during the forging process, making it easier for the 90-degree-rotated workpiece to be settled back in the appropriate position of the lower die, and it is also used to take out the forged workpiece. As shown in Figure 2, the octagonal angle of the longitudinal cross-section of the crevice was set to 70 degrees, and the angle of the horizontal guide crevice and the vertical guide crevice were also set to 14 degrees. The width W2 of the vertical crevice should be slightly larger than the width W1 (3 mm) of the horizontal crevice so that the workpiece that is rotated 90 degrees can be smoothly inserted into the lower die. Mechanical property can be evaluated by extracting a tensile test specimen with a diameter of 6 mm or less according to the ASTM: B557M-10 [19] standard’s Small-Size Specifics Professional to Standard. The biaxial alternate forging dies and experimental set-up are shown in Figure 4.

## 3. Results and Discussion

By conducting finite element analysis, the changes in the deformation behavior of the workpiece in the biaxial alternate forging were examined, and the amount of strain accumulated in the workpiece was analyzed according to the material properties. 

### 3.1. Finite Element Analysis Conditions

The commercial implicit finite element analysis software DEFORM v12.0.1 was used to simulate the biaxial alternating forging process. To reduce the computational time and increase the accuracy of analysis, the workpiece was modeled as only 1/8 considering the geometrical symmetry, as shown in Figure 5. However, to consider the rotation of the workpiece for each forging pass, a half model was used for the die, and the lower die was omitted because the shape of the upper and lower die was the same. For each forging pass, the workpiece was fixed, and the analysis was performed by rotating only the upper die by +90 and −90 degrees based on the central axis of the workpiece’s longitudinal direction. A four-node tetrahedral element was used for flexible automatic remeshing in the area where mesh deformation was excessive. The number of initial finite elements was set to about 180,000, and the size ratio of the elements was set to 1 to prevent internal elements from becoming coarse while reflecting the small round part of the crevice area of the die well during the deformation of the workpiece. The forging workpiece material was assumed to be a rigid–plastic material, and the die was assumed to be a rigid element. Friction was assumed to be shear friction, and 0.4 was applied as the m value.

### 3.2. Flow Stress for Material

The accumulation of effective strain within the workpiece during biaxial alternating forging is contingent upon the flow stress of the material. At room temperature, the flow stress of the material is mainly affected by the strain-hardening characteristics, but at high temperatures, recovery and recrystallization occur at the same time during the deformation process, and the effect of strain rate becomes more dominant than that of strain hardening [20]. Since this study deals only with biaxial alternating forging at room temperature, we analyzed the effect of the strain hardening characteristics on the deformation behavior of the material through finite element analysis. 

There are various hardening equations that represent the strain-hardening characteristics of materials [21]. In this study, Hollomon’s power-law equation, the simplest and most representative equation that approximates the strain-hardening behavior of flow stress in the form of an exponential function, was used. The power-law of the true stress–strain equation is described as follows [22]:$$\bar{\begin{array}{c} \sigma \end{array}}=K{\bar{\varepsilon }}^{n}$$

Figure 6 shows the changes in stress and strain curves depending on the strain-hardening exponent *n*. Depending on the value of the strain-hardening exponent, *n* = 0 and *n* = 1 represent the rigid–perfectly plastic and the elastic behaviors, respectively. To analyze the effect of the strain-hardening characteristics on the deformation of the biaxial alternate forged workpiece, the *K* value was fixed to an arbitrary value (594 MPa), and a finite element analysis was performed according to the change in the *n* value. Since it is judged that the *n* value does not exceed 0.6, the *n* values considered in the finite element analysis were 0, 0.2, 0.4, and 0.6. 

### 3.3. Finite Element Analysis Results

#### 3.3.1. Results of Deformation Analysis Depending on Strain-Hardening Exponent

Figure 7 shows the shapes of the deformed workpiece with an increasing number of forging passes of a material with an *n* value of 0.2 predicted through finite element analysis. As the forging pass increases, the shape of the guide flash of the forging workpiece becomes clear, and the cross-sectional shape of the center of the workpiece remains almost constant, as intended. Figure 8 shows the deformed shapes depending on the *n* values after the four forging passes. In the case of the rigid–perfectly plastic material with an *n* value of 0, it is confirmed that the development of the guide flash is the least, while that of the end protrusion is the most remarkable. It can be seen that as the *n* value increases, the guide flash develops uniformly, and the shape of the protrusion becomes gentle. Figure 9 shows the protruding lengths of the workpiece end for each forging pass. When the *n* value is 0, the protruding amount is almost linearly proportional to the forging passes, and above 0.2, it becomes larger in the beginning but decreases as the *n* value increases. This is attributed to the dispersion of deformation caused by the strain hardening. Figure 10 shows the distributions of the effective strain in the central section of the workpiece after four passes, and it can be seen that the effective strain concentrated in the core when the *n* value is 0 tends to disperse around as the *n* value increases. This trend can be found in Figure 11a, showing the distributions of the effective strain in the cross-section of the center of the workpiece depending on the *n* values. In particular, in Figure 11b describing the distribution of the metal flow lines, the metal flow lines are gradually distributed as the *n* value increases, indicating that the deformation is spreading well to the surroundings. These results suggest that the closer the *n* value is to 0, the more concentrated the deformation of the material in an X-shape, so it is difficult to propagate the deformation, but rather, in the center, the SPD such as ECAP can be obtained with just a few passes. In the graphs of Figure 10b–d, the positions corresponding to the diameters of the reduced section of the tensile test specimen of 4 mm and 6 mm are marked. A tensile test specimen with a diameter of 4 mm includes a deformed area under most *n* value conditions, so it is considered to exhibit relatively reliable mechanical characteristic values. In particular, if the *n* value is 0, it is recommended to use a tensile specimen with a diameter of the reduced section of 4 mm. 

#### 3.3.2. Estimations of Average Effective Strain Depending on Forged Passes

Figure 12 shows the changes in the maximum effective strain profiles of the center of the workpiece generated by biaxial alternating forging depending on the number of forging passes and the *n* values. However, as can be seen in Figure 10 and Figure 11, the distributions of the effective strain of the cross-section of the workpiece vary from location to location, so it is not appropriate to represent the strain generated in the workpiece through the maximum effective strain value. Therefore, only the area corresponding to the reduced section of the tensile test specimen was extracted from the analyzed model, and the average values of the effective strain depending on the *n* value and forging passes were calculated as shown in Figure 13. The average effective strain in the region with the reduced section diameter of 4 mm is higher than that of 6 mm because the inside of 4mm contains fewer areas with a lower effective strain than that of 6mm. Figure 13a shows that when the *n* value is 0, the average effective strain of 440% occurs within 4 mm in the diameter of the center of the workpiece after eight passes, and when the *n* value is 0.2, the average effective strain of 300% occurs in the same pass. This indicates that biaxial alternate forging can be used as a method of imparting SPD inside the material. In Figure 14, the average effective strain in the workpiece is presented in the form of maps depending on the number of forging passes and the *n* values. Based on those maps, if the *n* value of the material is determined, it is possible to estimate the average effective strains depending on the forging passes without conducting finite element analysis. To find more accurate values, the curves and polynomials in Figure 15 can be used. Figure 15 shows the relation between the average effective strain and the *n* value. The effective strain values depending on the *n* value for each pass were fitted to a third-order polynomial, and the values fitted were shown in Table 1. By substituting the *n* value of material into the *x* parameter of the polynomial, the average effective strain for each pass can be immediately calculated.

## 4. Biaxial Alternating Forging of 6061 Al Alloy

From the experiments on the 6061 aluminum alloy using biaxial alternating forging dies, we attempted to assess the practicality of biaxial alternating forging as an SPD method. The 6061 aluminum alloy used in the experiment was manufactured in the form of a billet with a diameter of Ø127 mm through continuous casting and was homogenized at 530 °C for 8 h to dissolve segregations. To produce the workpiece, a rod-shaped part with a diameter of Ø19 and length of 109 mm was extracted from the billet through machining, and both ends were rounded to 6 mm. In order to induce plane strain deformation in the axial direction of the workpiece, it is necessary to suppress axial elongation as much as possible. Therefore, in this experiment, the forging was performed under dry friction conditions without applying a lubricant. Figure 16 shows the shape of the material forged up to 8 passes through the biaxial alternating forging. As the number of forging passes increases, both ends of the workpieces gradually protrude, and cracking occurs from the 7 passes. On the other hand, no cracks were found inside the workpiece where hydrostatic pressure was applied during deformation. The results of the tensile test using tensile specimens with a reduced cross-section diameter of 6 mm taken from a workpiece according to ASTM standards are shown in Figure 17. As the forging pass increases, the tensile strength gradually increases. The elongation decreases rapidly in the beginning, but the decrease becomes more gradual as the forging is repeated. To calculate the average effective strain inside the workpiece, the polynomial expression in Table 1 was used. To determine the work hardening index of the 6061 aluminum alloy, the flow stress was derived through a compression test. At this time, the compression specimen was machined to Ø12 mm and 15 mm in height, and the compression test was conducted using a SHIMADZU Hydraulic Universal Testing Machine UH-1000 kNI (SHIMADZU Corp., Kyoto, Japan). The true stress–true strain obtained through the compression test and the curve fitting using Hollomon’s power-law equation are shown in Figure 18. The *K* value and the strain hardening exponent *n* were found to be 221.975 and 0.157789, respectively. This strain-hardening exponent value was substituted for the polynomial expression in Table 1 to calculate the average effective strain at the center of the workpiece according to the forging pass as shown in Table 2. It is confirmed that an average effective strain of 264% can be obtained after 8 passes, and also it can be seen that an SPD at a level that significantly exceeds the elongation of the raw material occurred inside the workpiece. Figure 19 displays the changes in tensile properties depending on the average effective strain mentioned above. The SPD behavior was observed that allows for a continuous increase in strength without a significant decrease in elongation except for a rapid reduction in elongation after forging with 1 pass.

Figure 20 shows the optical micrographs of the forged workpieces. While the microstructures of the U region with a low strain exhibited no significant difference from that before the forging (0 pass), the presence of twins in grains was observed in the high-strain regions, C and D. Even though a large strain of 264% after 8 passes was applied, no recrystallized structure was observed in the tissue photo, while the deformed grains and twins were found. It is considered that, in biaxial alternating forging performed at room temperature, grain refinement, which is one of the typical characteristics in the SPD process, is not observed, even when large strains are applied. It appears that the strain caused by biaxial alternating forging is absorbed through grain deformation due to dislocation growth and slip, as well as twin formation. It was reported that the grain refinement observed in the SPD process mainly occurs at temperatures above 0.5Tm, and it is attributed to dynamic recrystallization caused by strain energy accumulation inside the material [23,24]. Therefore, it is thought that the forging temperature in this study was insufficiently high enough to induce recrystallization. 

## 5. Conclusions

This study was conducted to investigate the practicability of biaxial alternate forging as a method to impose SPD to a 6061 aluminum alloy in cold working. Based on finite element analysis and experiments, some important conclusions are summarized as follows:As a result of the finite element analysis on biaxial alternate forging, it was confirmed that the strain distribution and amount of the effective strain accumulated inside the workpiece per forging pass varied depending on the strain hardening exponent, which changes the deformation behavior of the workpiece.The calculation of the strain distribution in the area within the diameter 4 mm of the reduced cross-section in the tensile specimen showed that the average effective strains after eight passes were approximately 440% and 300% when the strain-hardening exponent is 0 and 0.2, respectively, indicating that the biaxial alternate forging is suitable for an SPD process.The average effective strain of the 6061 aluminum alloy within the diameter of 6 mm from the core of the workpiece after eight passes of forging was found to be approximately 264%. This indicates that it can impose a large strain that significantly exceeds the elongation of the raw material.As a result of the tensile test on the forged 6061 aluminum alloy workpieces, the strength gradually increased without a significant reduction in elongation after two passes. The optical microstructures in the center of the forged workpieces showed the distribution of deformed grains and twins with no recrystallization throughout all the forging passes.

## Figures and Tables

**Figure 1 materials-17-00968-f001:**
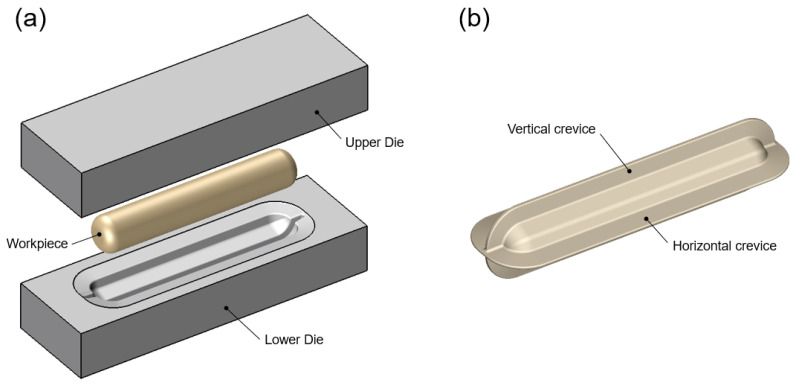
Schematic 3D views of (**a**) the tools and the workpiece for biaxial alternate forging and (**b**) the die cavity shape.

**Figure 2 materials-17-00968-f002:**
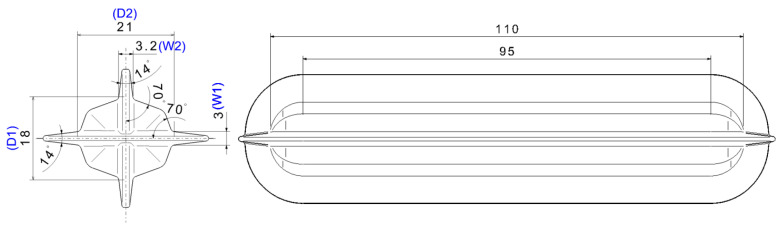
Dimensions of the die cavity.

**Figure 3 materials-17-00968-f003:**
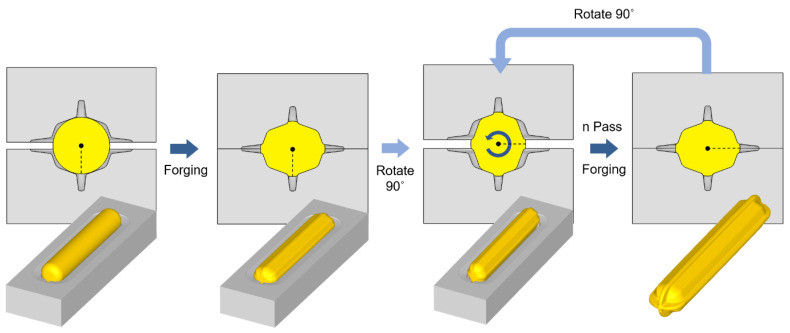
Schematic views of biaxial alternate forging process using octagonal rod-shaped dies.

**Figure 4 materials-17-00968-f004:**
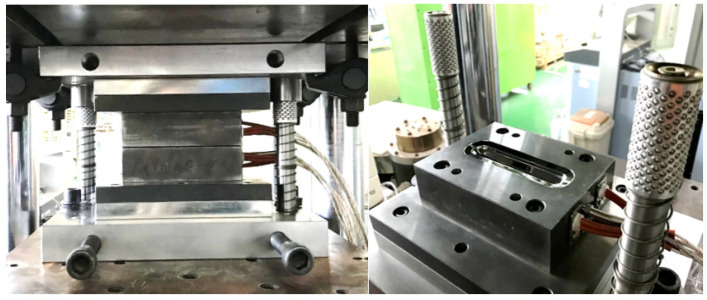
Biaxial alternate forging dies and experimental set-up.

**Figure 5 materials-17-00968-f005:**
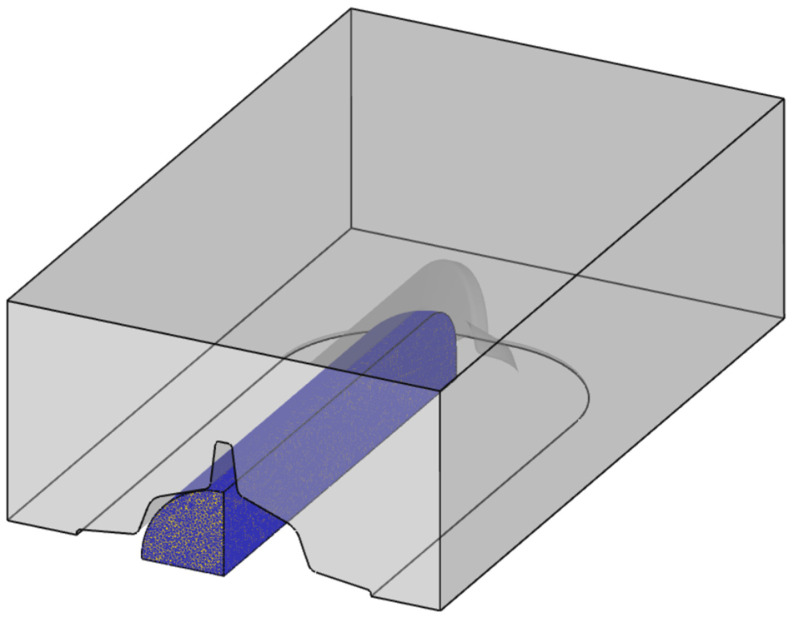
Finite element modeling for simulation of biaxial alternate forging.

**Figure 6 materials-17-00968-f006:**
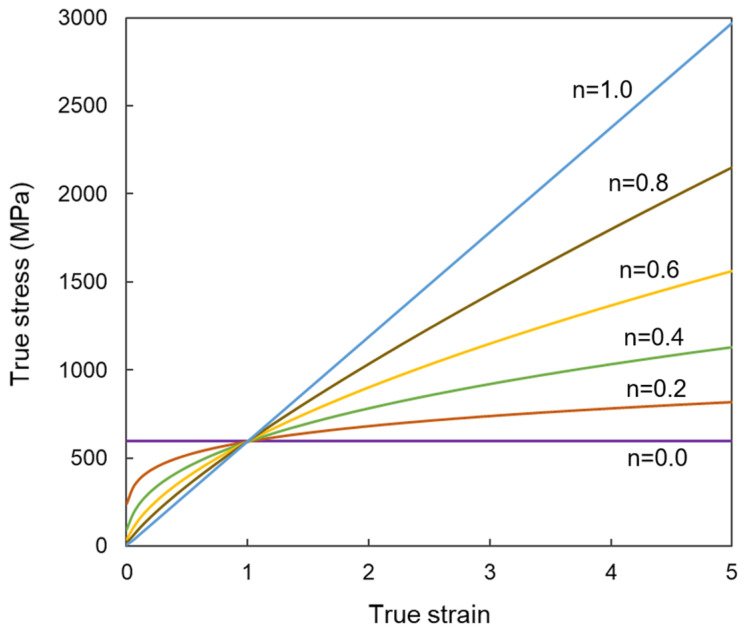
True stress–strain curves fitted to power-law depending on *n* value.

**Figure 7 materials-17-00968-f007:**
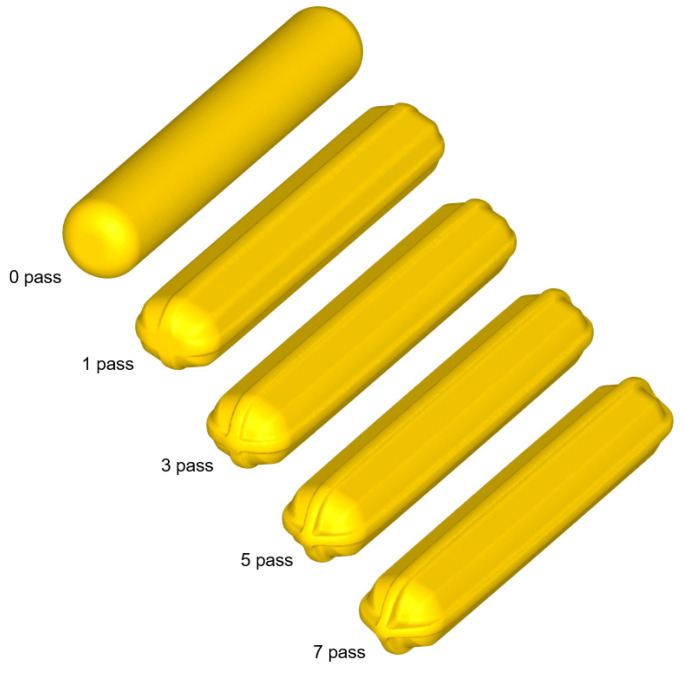
Shape of the deformed workpiece with increasing number of forging passes.

**Figure 8 materials-17-00968-f008:**
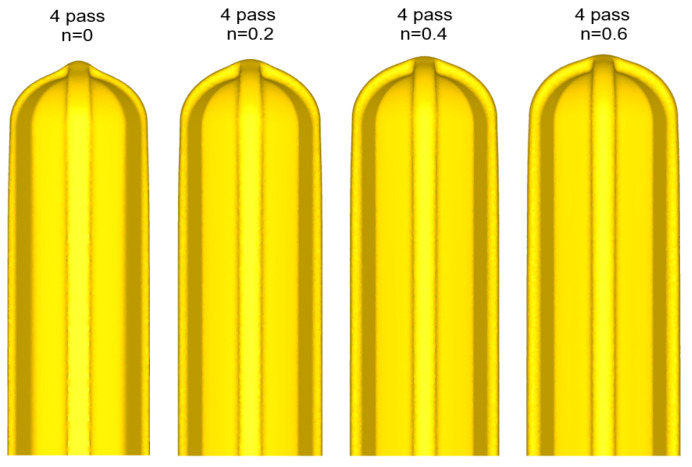
Comparison of deformed shapes depending on strain-hardening exponent after the four forging passes.

**Figure 9 materials-17-00968-f009:**
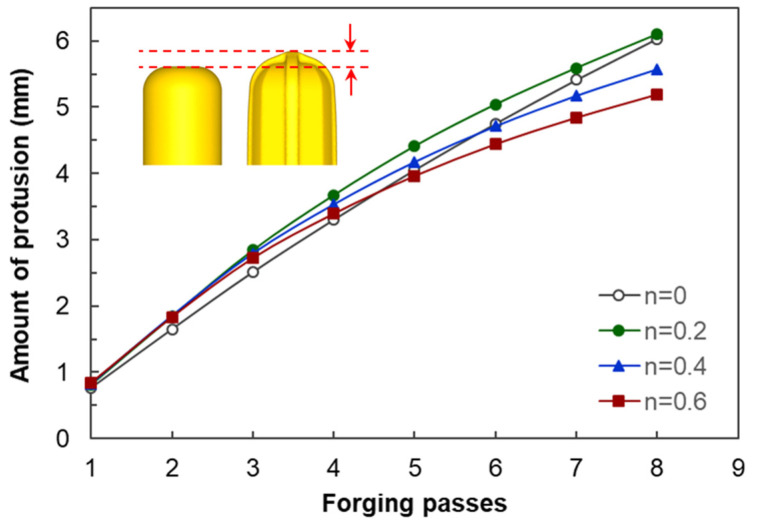
Variation in amount of protrusion depending on the number of forging passes and the *n* values.

**Figure 10 materials-17-00968-f010:**
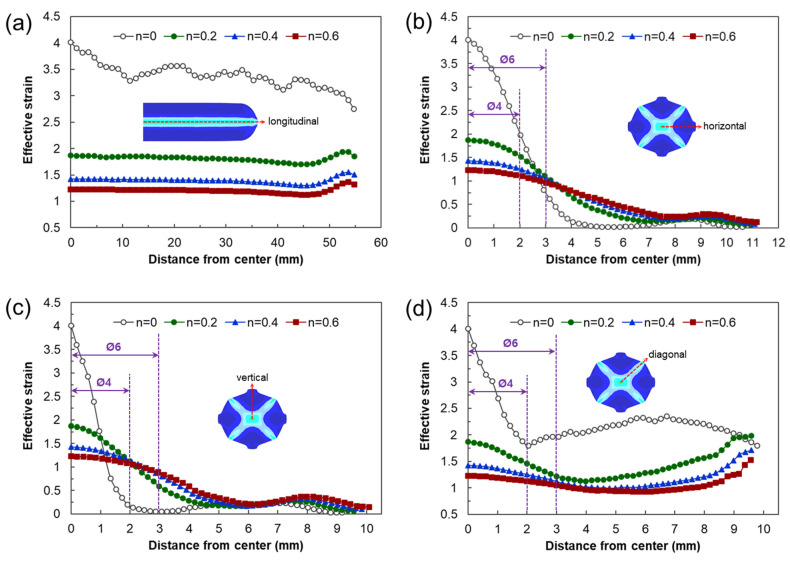
Effective strain profiles along the (**a**) horizontal, (**b**) diagonal, and (**c**) vertical directions, depending on the *n* values. (**d**) Difference in effective strains depending on the directions in the cross-section after the four forging passes.

**Figure 11 materials-17-00968-f011:**
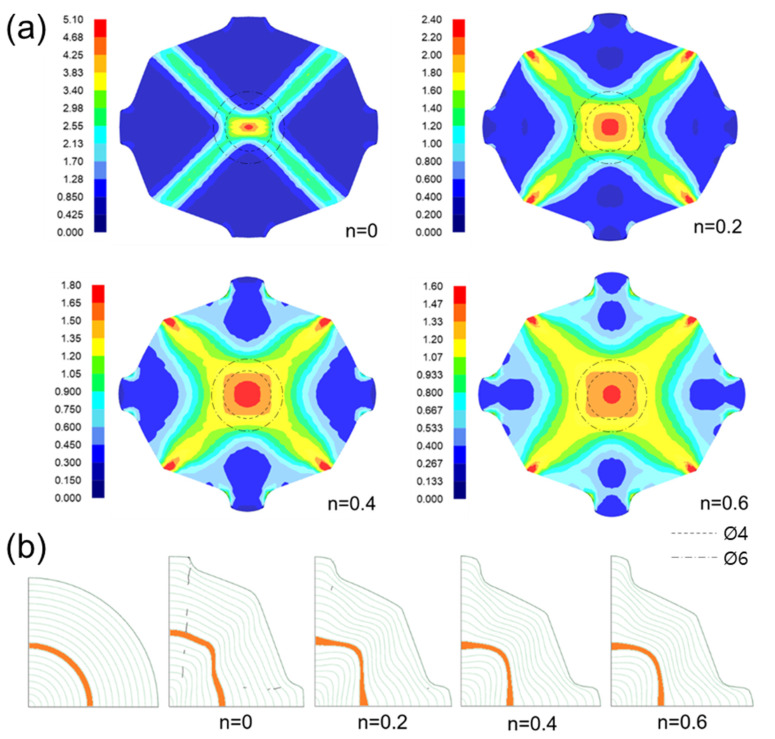
(**a**) Effective strain distributions and (**b**) metal flow lines (green lines) in the cross-section of workpieces after the five forging passes depending on the *n* values.

**Figure 12 materials-17-00968-f012:**
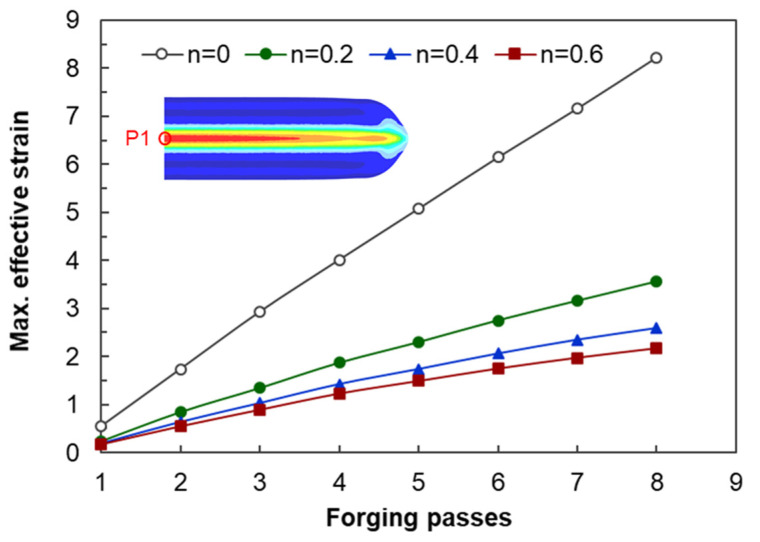
Maximum effective strain profiles depending on the number of forging passes and the *n* values.

**Figure 13 materials-17-00968-f013:**
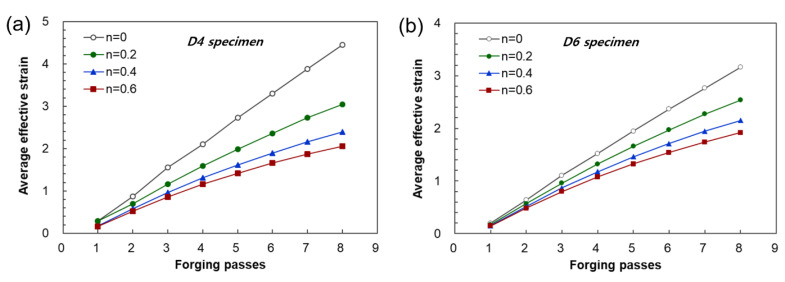
Average effective strain profiles of measuring areas for tensile tests using (**a**) D4 specimen and (**b**) D6 specimen depending on the number of forging passes and the *n* values.

**Figure 14 materials-17-00968-f014:**
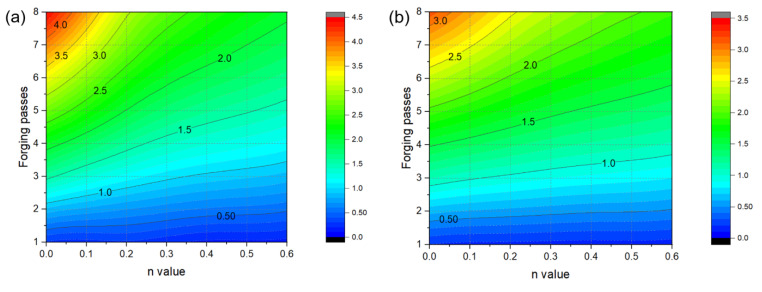
Average effective strain map of measuring areas for tensile tests using (**a**) D4 specimen and (**b**) D6 specimen.

**Figure 15 materials-17-00968-f015:**
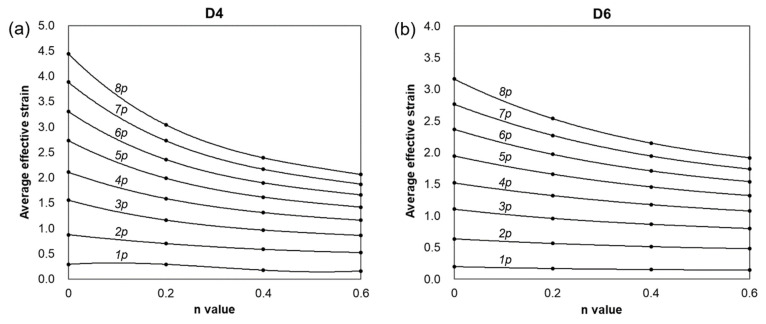
Estimation of effective strain values in measuring areas for tensile tests using (**a**) D4 specimen and (**b**) D6 specimen.

**Figure 16 materials-17-00968-f016:**
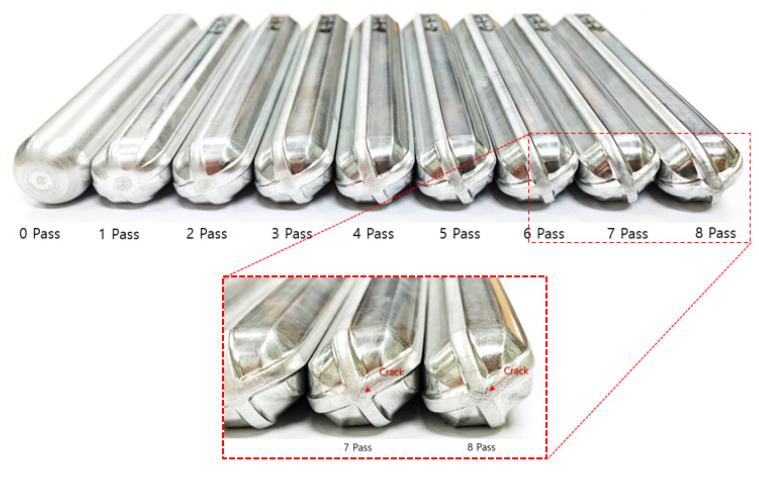
Appearance of workpieces depending on the number of forging passes.

**Figure 17 materials-17-00968-f017:**
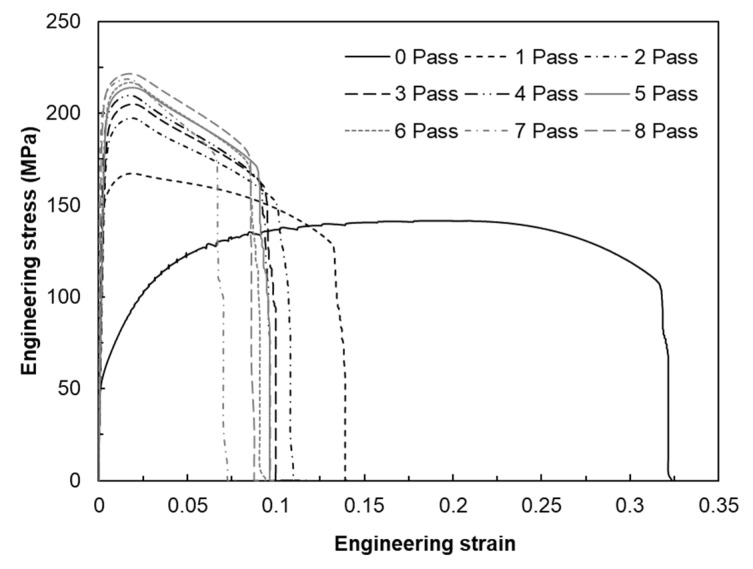
Change in tensile curves of 6061 aluminum workpieces depending on the number of forging passes.

**Figure 18 materials-17-00968-f018:**
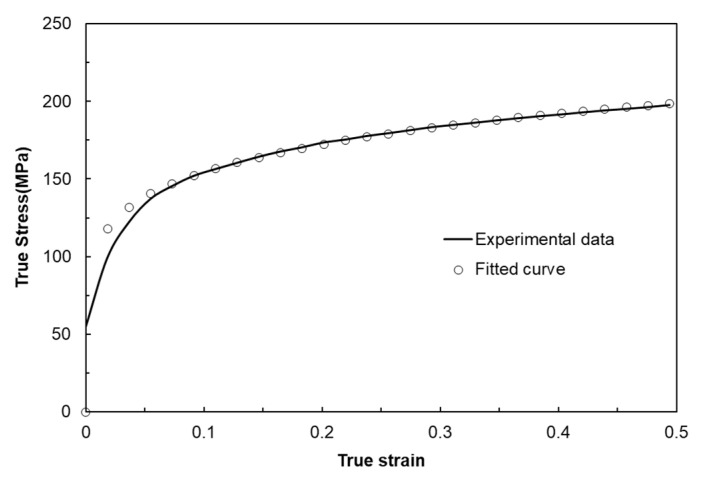
True stress–strain curve of 6061 aluminum obtained from compression test at room temperature and curve fitting result using power-law equation.

**Figure 19 materials-17-00968-f019:**
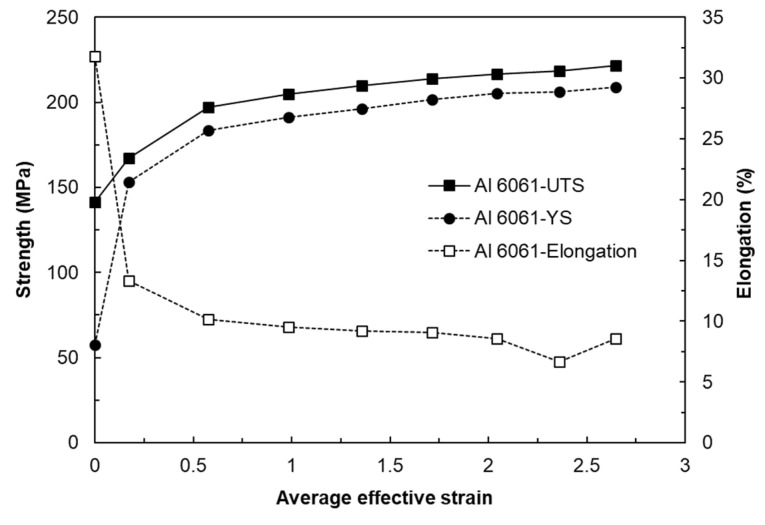
Change in tensile properties of 6061 aluminum workpieces depending on the number of forging passes.

**Figure 20 materials-17-00968-f020:**
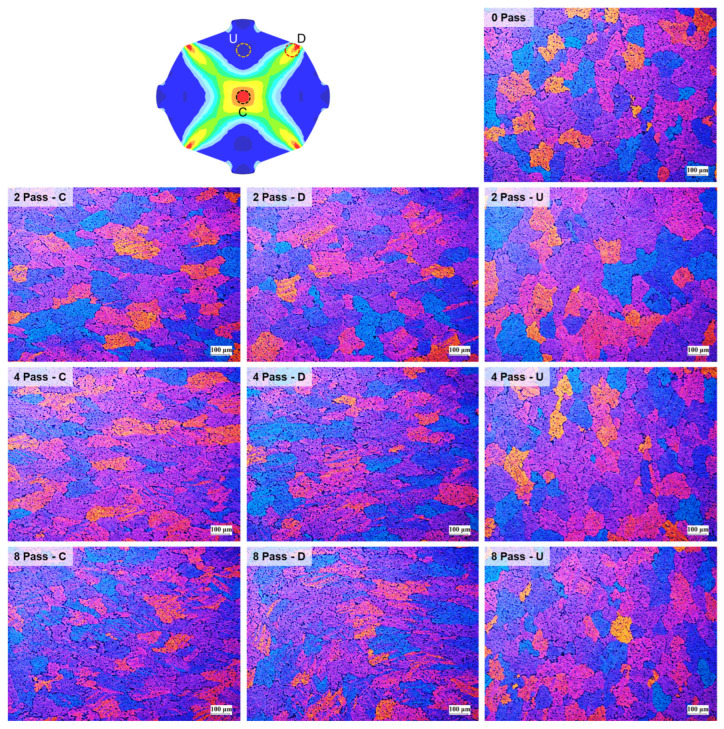
OM micrographs of forged 6061 aluminum workpiece depending on the number of forging passes and position on the cross-section.

**Table 1 materials-17-00968-t001:** The polynomial fitted values of effective strain values in measuring areas for tensile tests.

Forging Passes	${Ax}^{3}+Bx^{2}+Cx+D$							
	D4 Specimen				D6 Specimen			
	A	B	C	D	A	B	C	D
1	4.4994	−4.1337	0.6468	0.2887	−0.1035	0.2315	−0.1900	0.1977
2	−0.1765	0.8638	−1.0399	0.8733	−0.1513	0.3860	−0.4381	0.6375
3	−2.2025	3.7803	−2.6383	1.5559	−0.5462	0.9872	−0.9052	1.1059
4	−2.3728	4.4439	−3.3914	2.1069	−0.2585	0.8547	−1.1574	1.5202
5	−4.0397	7.0479	−4.9670	2.7320	−0.5020	1.4089	−1.7096	1.9485
6	−5.5281	9.3960	−6.3917	3.3037	−0.8550	2.1942	−2.3873	2.3689
7	−6.7134	11.422	−7.7897	3.8821	−0.9281	2.6342	−2.9525	2.7644
8	−8.9278	14.736	−9.6048	4.4453	−1.7251	4.0135	−3.8606	3.1622

**Table 2 materials-17-00968-t002:** The estimated average effective strain of 6061 aluminum alloy depending on the number of forging passes calculated from the polynomial equations of Table 1 (D6 specimen).

Pass	1	2	3	4	5	6	7	8
Avg. effective strain (%)	17.3	57.7	98.6	135.8	171.2	204.3	236.0	264.6

## Data Availability

Data are contained within the article.

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
