# Peer review of "Deformation Behavior and Microstructure of 6061 Aluminum Alloy Processed by Severe Plastic Deformation Using Biaxial Alternate Forging"

_materials, 2024, doi:10.3390/ma17050968_

Round 1
Reviewer 1 Report
Comments and Suggestions for Authors
Points in Favor:
1. The biaxial alternate forging technique is an interesting alternative to other SPD processes that have been practiced for the past couple of decades, and has the potential of producing bulk SPD material, in contrast to two-dimensional or sheet product that is made by processes such as HPT or ARB.
2. The use of octagonal dies with “flash cavities” in the vertical and horizontal directions is a clever way to impose compressive deformation in two perpendicular directions.
3. Experimental verification of the finite element modeling of the deformation process shows potential for the process, in that the FEA predicts a product that is similar to the experimental results.
Unfavorable Points
1. The finite element simulations show shear bands with high effective strain along the 45o directions and a high strain along the longitudinal axis of the work-piece. The high average strain associated with the definition of SPD is reached to only in the region of about 4 to 6 mm diameter in the middle of a workpiece, which has a starting diameter of around 19 mm. The SPD region is therefore only about 10-12% of the volume of the workpiece.
2. The FEA seems to have been done with assumption that the strain hardening exponent n is constant for each set of 8 passes. However, it is clear from the experimental results shown in Figure 16 that the strain hardening exponent changes with each pass, with an initially high n value in the 0-pass material and gradually decreasing up to the 8-pass material. Could the FEA results be improved by taking this change in n value with strain (or pass number)?
3. Though the polynomial fit to the change in effective strain with n for different passes appears to be effective, it is only a curve-fitting exercise that provides little any insight about the physical phenomena that could be occurring at the microstructural level.
Suggested revisions
4. The manuscript has been marked with highlights and comments at multiple locations that have additional questions or need clarifications.
5. The optical micrographs in Figure 19 are quite intriguing. It is not clear where in the cross-section of the workpiece they were taken. Though the starting microstructure shows equiaxed grains, the micrographs after 2,4, and 8 passes are remarkably similar – grain that are elongated at 45o to the vertical direction in the micrograph. There does not seem to be any grain refinement with increasing strain as one would expect in SPD, or significant deformation that would result in decreased ductility and higher strength, as one would expect with cold-working by conventional deformation processes like rolling. This observation alone needs additional investigation.

Author Response
(Points in Favor)
- The biaxial alternate forging technique is an interesting alternative to other SPD processes that have been practiced for the past couple of decades, and has the potential of producing bulk SPD material, in contrast to two-dimensional or sheet product that is made by processes such as HPT or ARB.
- The use of octagonal dies with “flash cavities” in the vertical and horizontal directions is a clever way to impose compressive deformation in two perpendicular directions.
- Experimental verification of the finite element modeling of the deformation process shows potential for the process, in that the FEA predicts a product that is similar to the experimental results.
Reviewer #1: The paper was improved after the modification. If possible, please refer the following things:
Please modify the numbers on ordinate, e.g., 106 -> 10^6.
Please add the scale bar on Fig. 4.- Authors’ answer
: Authors would like to thank the reviewer for the favorable comments.
(Unfavorable Points)
- The finite element simulations show shear bands with high effective strain along the 45o directions and a high strain along the longitudinal axis of the work-piece. The high average strain associated with the definition of SPD is reached to only in the region of about 4 to 6 mm diameter in the middle of a workpiece, which has a starting diameter of around 19 mm. The SPD region is therefore only about 10-12% of the volume of the workpiece.
- Authors’ answer
: Rather than imposing SPD to the entire part of material, BAF test aims to provide an effective method of applying severe plastic strain to the material and to examine various phenomena that occur during the material's deformation process. And also, microstructure analysis and tensile test have been conducted only on the areas where SPD occurred.
- The FEA seems to have been done with assumption that the strain hardening exponent n is constant for each set of 8 passes. However, it is clear from the experimental results shown in Figure 16 that the strain hardening exponent changes with each pass, with an initially high n value in the 0-pass material and gradually decreasing up to the 8-pass material. Could the FEA results be improved by taking this change in n value with strain (or pass number)?
- Authors’ answer
: Since the strain hardening exponent is a factor of the degree to which a material strengthens depending on strain, it is also commonly used in strain accumulation, as in this study. In FEA, the strain is accumulated depending on forging pass, and in the next pass, the material deforms according to the strain hardening curve after the strain occurs, so it is said that the strain hardening exponent changes in each pass. Similarly, the change in work hardening in Figure 17 is caused by strain after the forging pass. Although a forging analysis based on a constant strain hardening curve cannot perfectly reflect the change in strain hardening exponent during the experimental process, it is expected to simulate similar trends in the experiment. In addition, even if the strain hardening exponent that changes during the experiment is measured through a tensile test, since it is the average value for the reduced cross-section of tensile specimen, it cannot reflect the variation in strain depending on the location, so the analysis is expected to be more inaccurate.
- Though the polynomial fit to the change in effective strain with n for different passes appears to be effective, it is only a curve-fitting exercise that provides little any insight about the physical phenomena that could be occurring at the microstructural level.
Reviewer #1: The paper was improved after the modification. If possible, please refer the following things:
Please modify the numbers on ordinate, e.g., 106 -> 10^6.
Please add the scale bar on Fig. 4.- Authors’ answer
: The polynomial fit results are values for quantitatively comparing the accumulated amount of SPD imposed inside the specimen in the BAF test for each forging pass. Therefore, it cannot provide intuition about physical phenomena at the microstructural level.
(Suggested Revisions)
- The manuscript has been marked with highlights and comments at multiple locations that have additional questions or need clarifications.
- Authors’ answer
: Authors would like to thank the reviewer for the thorough review. The authors have made corrections according to all comments the review put in the manuscript and responses in the manuscript file.
- The optical micrographs in Figure 19 are quite intriguing. It is not clear where in the cross-section of the workpiece they were taken. Though the starting microstructure shows equiaxed grains, the micrographs after 2,4, and 8 passes are remarkably similar – grain that are elongated at 45o to the vertical direction in the micrograph. There does not seem to be any grain refinement with increasing strain as one would expect in SPD, or significant deformation that would result in decreased ductility and higher strength, as one would expect with cold-working by conventional deformation processes like rolling. This observation alone needs additional investigation.
- Authors’ answer
: The observed area is for the core of specimen. The points the reviewer mentioned are the characteristics of the BAF conducted in cold worked condition. Although SPD was imposed to the core of material, there was no visible change in the microstructure. Because dynamic recrystallization, which induces grain refinement in SPD, is proportional to temperature, no significant change in grain structures was obtained in this study.
(Answers to comments from reviewer 1 in manuscript)
*The following numbers refer to the line numbers with the comments in the file provided by the reviewer 1.
92 : Correction has been made.
187 : Metal flow was expressed as green line in Figure 11(b). Green lines have been shown in the caption of Figure 11.
197 : Correction has been made.
198 : The n value was put as a curve. If a deformation occurs in the simulation, the curve after that will be followed in the next pass. Therefore, similarly to the experiment, it can be considered the n value changes at each pass.
203 : When n = 0, the dispersion of strain is not smooth and local deformation occurs. Therefore, at the beginning of the pass, the deformation is mainly concentrated in forming the guide flash and a protrusion at the ends does not occur significantly. However, when n = 0, as the forging pass progresses, the strain in the core becomes prominent as shown in Figure 11, leading to the protrusion at the ends,. This trend is also shown in Figure 8.
213 : Modification has been made.
212 : This was mentioned earlier in the explanation of Figure 11(b).
222 : After drawing and meshing the cylinder-shaped geometry corresponding to the reduced cross-section of the tensile specimen, the effective strain formed in the workpiece was interpolated and added to that portion. Then, the effective strain values at each element was extracted and their average value was calculated.
252 : It was a simple way of fitting equation that could most closely represent the change in average effective strain.
279 : More explanation has been made.
287 : The cracking occurred only in the flash and not inside.
289 : It is believed that the specimen had slight defects. It does not seem reasonable to consider this as a result of another phenomenon, compared to the results of previous two studies that examined BAF on two types of Al-Mg alloys (Currently, Ref. 14, 15)
292 : In experimental data, the elastic deformation was removed. The mark has been modified to reduce confusion.
298 : More explanation has been made. Explanation in Line 238~239 (before modification) is helpful for understanding.
301 : Modification has been made.
303 : The word has been corrected to micrographs.
313 : The microstructures were taken from the core of workpieces. Because the workpieces are rotated 90 degrees with each forging pass, the shape of grains in the even numbers of forging passes would have no significant difference from those before forging. Unless a dynamic recrystallization occurs, no significant change in the shape of the grains may be found.
314 : Modification has been made.

Reviewer 2 Report
Comments and Suggestions for Authors
The article focuses on FE-modeling the workpiece’s deformation process using alternating forging. The direction of the research is interesting, but I have a lot of criticism for the paper. A major revision is required before the article can be accepted for publication.
1. The authors used the alternating forging technique in their study. In general terms, this technique is similar to the well-known multi-step isothermal forging technique, which should be mentioned through the relevant references [Progress in Materials Science 45 (2000) 103–189; Materials Science & Engineering A 709 (2018) 330–338].
2. Line 34-58. The authors provide many references related to the processing of aluminum alloys by severe plastic deformation methods. However, only listing these references without analyzing the results obtained is uninformative. A more in-depth analysis of literature data is needed in relation to your research.
3. The authors note an improvement in strength of Al-alloy without significant degradation of elongation. However, this is completely false. The elongation decreased many times over. Uniform elongation, which most adequately characterizes the ductility of the material, has decreased especially noticeably. This point needs to be reconsidered.
4. Figure 5. ‘True strain’ instead of ‘Ture strain’ on the X-axis
5. Some explanations of the FE-modeling methodology are necessary. How was the whole workpiece in Figure 6 obtained if 1/4 of the model was used? What was the workpiece model’s diameter? How is it related to the experimental workpiece’s diameter?
6. The obtained modeling results depend on the strain hardening coefficient. How common are they? Do they depend on the type of workpiece’s material?
7. Why did you use the 6-mm-diameter tensile specimens, if the modeling showed that the 4-mm-diameter specimens are most adequate?
8. Line 259. The word ‘diameter’ seems to be missing.
Author Response
The article focuses on FE-modeling the workpiece’s deformation process using alternating forging. The direction of the research is interesting, but I have a lot of criticism for the paper. A major revision is required before the article can be accepted for publication.
- Authors’ answer
: Authors would like to thank the reviewer for the insightful review. The authors have revised the manuscript according to the reviewers' comments.
Reviewer #2: This revised manuscript reads well. I only had a few minor revisions, which I marked in the attached document.
The manuscript provides a good support for the presented ideas, and so I would rate this as a "minor" revision.
- The authors used the alternating forging technique in their study. In general terms, this technique is similar to the well-known multi-step isothermal forging technique, which should be mentioned through the relevant references [Progress in Materials Science 45 (2000) 103–189; Materials Science & Engineering A 709 (2018) 330–338].
- Authors’ answer
: Authors would like to thank the reviewer for recommending a valuable reference. This article has been mentioned and cited in our manuscript.
- Line 34-58. The authors provide many references related to the processing of aluminum alloys by severe plastic deformation methods. However, only listing these references without analyzing the results obtained is uninformative. A more in-depth analysis of literature data is needed in relation to your research.
- Authors’ answer
: Authors have added more explanation for the literatures.
- The authors note an improvement in strength of Al-alloy without significant degradation of elongation. However, this is completely false. The elongation decreased many times over. Uniform elongation, which most adequately characterizes the ductility of the material, has decreased especially noticeably. This point needs to be reconsidered.
- Authors’ answer
: The authors focused on the trend after the rapid decrease in elongation with one pass of forging. The modification of explanation has been made.
- Figure 5. ‘True strain’ instead of ‘Ture strain’ on the X-axis
- Authors’ answer
: Authors would like to thank the reviewer for the thorough review. Correction has been made.
- Some explanations of the FE-modeling methodology are necessary. How was the whole workpiece in Figure 6 obtained if 1/4 of the model was used? What was the workpiece model’s diameter? How is it related to the experimental workpiece’s diameter?
- Authors’ answer
: Since the model was minimized based on the symmetry of shape and deformation behavior of workpiece, the shape shown in Figure 6 can be obtained by symmetrically duplicating it using the mirror symmetry definition function in the post processing. The identical workpiece was examined in both experiments and analyzes. Please see the dimension of workpiece mentioned in Chapter 2.
- The obtained modeling results depend on the strain hardening coefficient. How common are they? Do they depend on the type of workpiece’s material?
- Authors’ answer
: The Power-law equation can be said to be the best-known model representing the flow stress of materials at room temperature, and the deformation behavior of actual materials at room temperature is affected by the strain hardening coefficient. If the material undergoes a pre-strain, the equation may be slightly different. On the other hand, the pre-strain was not considered in this study. In this case, the strain hardening coefficient varies depending on the workpiece’s material.
- Why did you use the 6-mm-diameter tensile specimens, if the modeling showed that the 4-mm-diameter specimens are most adequate?
- Authors’ answer
: The analysis didn’t indicate that the 4mm-diameter is more suitable, but rather that the 4mm-diameter contains less unstrained area than the 6mm-diameter. Since 6061 aluminum has a strain hardening exponent of 0.157789, it was considered that 6mm-diameter tensile specimen almost had no unstrained area. The 6mm-diameter tensile specimen was chosen to evaluate the mechanical properties of a relatively large area of forged workpiece.
- Line 259(261). The word ‘diameter’ seems to be missing.
- Authors’ answer
: Correction has been made.

Round 2
Reviewer 2 Report
Comments and Suggestions for Authors
I am satisfied with the authors' answers. The manuscript can be published.
P.S.: Please note that in Figure 6 the X-axis is still 'ture strain' instead of 'true strain'
Author Response
Authors would like to thank the reviewer for the meticulous comment. Correction has been made.